Microgeographic maladaptive performance and deme depression in response to roads and runoff

Brady Steven P. brady.steven@gmail.com
School of Forestry & Environmental Studies, Yale University , New Haven, CT , USA
Sorci Gabriele
Electronic publication date: 2013 Sep 17
Publication date: 2013
Volume: 1
Electronic Location ID: e163
Received 2013 May 10; Accepted 2013 Aug 28
Copyright: © 2013 Brady
Copyright year: 2013
Copyright holder: Brady
License: This is an open access article distributed under the terms of the Creative Commons Attribution License, which permits unrestricted use, distribution, and reproduction in any medium, provided the original author and source are credited.
License URL: https://creativecommons.org/licenses/by/3.0/

Keywords: Microgeographic differentiation, Adaptation and maladaptation, Contemporary evolution, Road salt, Conservation, Roads and runoff

Funding: National Science Foundation grant DEB 1011335 Mianus River Gorge Preserve Research Assistantship Program Yale Carpenter/Sperry/Mellon fund Yale Institute for Biospheric Studies Hixon Center for Urban Ecology American Museum of Natural History Theodore Roosevelt Fund Federated Garden Club of Connecticut This research was supported by funding from the National Science Foundation grant DEB 1011335, the Mianus River Gorge Preserve Research Assistantship Program, the Yale Carpenter/Sperry/Mellon fund, the Yale Institute for Biospheric Studies, the Hixon Center for Urban Ecology, the American Museum of Natural History Theodore Roosevelt Fund, and the Federated Garden Club of Connecticut. The funders had no role in study design, data collection and analysis, decision to publish, or preparation of the manuscript.

==============================
Despite theoretical understanding and empirical detection of local adaptation in natural environments, our knowledge of such divergence in fragmented habitats remains limited, especially in the context of microgeographic spatial scales and contemporary time scales. I used a combination of reciprocal transplant and common garden exposure experiments to evaluate potential microgeographic divergence in a pool-breeding amphibian occupying a landscape fragmented by roads. As indicated by reduced rates of survival and increased rates of malformation, I found evidence for maladaptation in road adjacent populations. This response is in direct counterpoint to recently described local adaption by a cohabiting species of amphibian. These results suggest that while divergence might commonly follow habitat modification, the direction of its outcome cannot be generalized even in identical habitats. Further, maladaptive responses can be associated with a more generalized depression effect that transcends the local environment. Alongside recent reports, these results suggest that maladaptive responses may be an emerging consequence of human-induced environmental change. Thus future studies should carefully consider the population unit as a key level for inference.

Introduction

Identifying informative scales and levels of biological organization for investigation has long been regarded as a critical challenge in ecology, limiting our ability to describe variation in the natural world (Levin, 1992). Population level approaches—which are bringing to light profound variation that occurs across small spatial and temporal scales (Pigliucci, 2005)—offer a promising advancement on this challenge. Variation observed at this level can reveal dramatic differences among local populations, even across microgeographic spatial scales (Skelly, 2004; Urban, 2010). For example, spatially heterogeneous selection pressures coupled with heritable phenotypic variation has often been found to result in local adaptation (Hereford, 2009; Kawecki & Ebert, 2004). Evidenced by a specific form of gene-by-environment interaction (G × E), local adaptation results when the local population has higher fitness than a foreign population measured within the local environment (Kawecki & Ebert, 2004). However, gene flow and genetic drift can constrain local adaptation, rendering populations equivalently adapted between environments, or even maladapted to their local environment (Crespi, 2000; Falk et al., 2012; Stearns & Sage, 1980; Urban, 2006). Though evident in wild populations (Moore & Hendry, 2009), such empirical examples of population level maladaptation are rare (Spitzer, 2006). Rather, most examples of maladaptation in general describe reduced fitness across higher levels of organization, typically in host-parasite or host-pathogen systems, where coevolution can generate reciprocating dynamics of maladaptation and adaptation (Thompson, 1999).

In human-altered contexts, where novel selection pressures and reduced gene flow can yield adaptive responses (Brady, 2012) and/or genetic drift (Delaney, Riley & Fisher, 2010), and the rates of phenotypic change are high relative to undisturbed settings (Darimont et al., 2009; Hendry, Farrugia & Kinnison, 2008), the potential for population divergence should be elevated. Given this increased potential, population level approaches should be viewed as essential for fulfilling conservation imperatives aimed at understanding the long-term fates of populations influenced by human activities. Here, I report the population specific responses of a pool-breeding amphibian (the wood frog, Rana sylvatica = Lithobates sylvaticus) occupying a landscape fragmented by roads. Given the suite of novel selection pressures associated with road fragmentation and runoff—along with previously documented contemporary local adaptation to temperature by the wood frog in this landscape (Skelly, 2004)—I expected that road adjacent populations of this species would be locally adapted to the roadside environment. In particular, I expected road runoff contaminants—especially road salt (Brady, 2012)—to act as the selective pressure conducing local adaptation in these populations. This outcome seems especially likely in light of recent evidence of local adaptation to roads by another amphibian (the spotted salamander, Ambystoma maculatum) breeding and dwelling in the same locations (Brady, 2012).

I hypothesized (1) that the roadside environment would induce negative effects on the aquatic life stages of the wood frog, and (2) that within a common roadside environment, roadside populations would outperform woodland populations (i.e., the local adaptation hypothesis). I used a field-based reciprocal transplant experiment (Figs. 1 and S1) coupled with a series of lab-based exposure experiments to test these hypotheses across a total of 10 populations (five roadside, five woodland). Each of these populations occupied a distinct seasonal pool, which comprises natural, typical breeding habitat. To evaluate demographic consequences of roads in this system, I surveyed wild population size and female reproductive effort. Finally, to investigate potential mechanisms by which road proximity influences wood frog performance, I characterized the environment at each pool by estimating a suite of abiotic variables associated with amphibian distribution and performance (Wellborn, Skelly & Werner, 1996). I expected that these characteristics might vary with road proximity, and thus act as potential agents of natural selection in roadside pools.

Figure 1 Population locations and reciprocal transplant design.

Locations of the 10 pools comprising transplants are shown. Asterisk on inset indicates approximate study location in northeastern Connecticut, USA. Like symbols indicate paired pools (red = roadside; blue = woodland). Interstate highway (I-84) and on/off-ramp infrastructure is indicated in yellow. Primary roads are heavily shaded, while secondary and unpaved roads are lightly shaded. Bar graph shows average specific conductance (µS; ±1 SE) in the woodland and roadside pools. From among a suite of abiotic variables, I found that pools differed only with respect to specific conductance, which indicates elevated ionic concentration, reflecting the increased presence of chloride ions in roadside pools that originate from road salt runoff.

Materials and Methods

Natural history and site selection

The wood frog is widely distributed throughout eastern North America and much of Canada, with a range extending from within the Arctic Circle to the southeastern United States. Within the study region, populations of this species breed in late March or early April, when adults migrate from upland terrestrial habitat into ephemeral wetlands to reproduce. The wood frog is an explosive breeder, with the majority of oviposition for a given population occurring in a single night (Berven & Grudzien, 1990). Populations in this study region typically breed synchronously within several days of one another, when each female oviposits one egg mass containing approximately 800 eggs per mass. Embryos develop over two-three weeks before hatching, and continue to develop as aquatic larvae throughout spring and early summer until they metamorphose into terrestrial juveniles.

My method for selecting populations for this study is previously reported in a reciprocal transplant study that indicated local adaptation to roads in the spotted salamander (Brady, 2012). Briefly, I selected the five roadside pools with the highest conductivity (µS) values (an indicator of road salt runoff) and with a breeding population comprising at least 10 wood frog families. For each roadside pool, a nearby woodland pool was then selected to minimize confounding effects. Specifically, I selected woodland pools based on the criterion of maximizing similarity in the abiotic conditions of canopy cover, pool size, and the presence and type of emergent vegetation. Reciprocal transplants (described below) were conducted within each of these five pool pairs (see Figs. 1 and S1 for overview and schematic). Eight of the 10 pools selected during the aforementioned spotted salamander study (Brady, 2012) were also selected here; the remaining two pools did not have breeding populations of the wood frog and thus two different pools were selected for the present study. Inference from these overlapping populations provides an especially strong basis for comparison of responses between two amphibians occupying identical habitats. When referring to the full complement of roadside or woodland populations, I use the term ‘deme’. Thus, ‘roadside deme’ refers collectively to the five roadside populations while ‘woodland deme’ refers collectively to the five woodland populations.

Reciprocal transplant experiment

In spring 2008, I collected from each of 10 pools (five roadside, five woodland) a subset of embryos from 10 egg masses less than 36 h old. To ensure collection within this timeframe, pools were monitored daily as the breeding season approached. Roadside pools were located <10 m from a paved road while woodland pools were located >200 m from the nearest paved road and at least 100 m from the nearest dirt road (Fig. 1). Dirt roads in this region typically see low traffic use and do not receive winter deicing salts. While roadside egg masses appeared somewhat more compact than woodland egg masses (Karraker & Gibbs, 2011), there were no conspicuous differences in the embryos themselves between sites, and selection of embryos from within each egg mass was haphazard. Thus, any potential carryover effect of the <36 h exposure did not bias the selection of the subset of embryos employed in the experiment. The consequences of such potential effects are elaborated upon in the Discussion. Each egg mass was collected whole in a plastic 710 ml container. From each egg mass, I separated out two clusters of embryos. Each cluster was photographed and stocked into one of five experimental enclosures in the origin pool and the transplant pool (Fig. S1). (Details of the experimental enclosures are described elsewhere (Brady, 2012) and an image can be found in Fig. S1.) For a subset of egg masses, this procedure occurred in the field. However, heavy rains impeded photography; remaining egg masses were instead transported to the lab on ice for processing, stored overnight in an 8°C incubator, and returned for stocking the following day. Each pool contained five experimental blocks. I targeted stocking approximately 50 embryos from each of two egg masses per enclosure, yielding one unique pairing of egg masses (hereafter “clutch pair”) that was replicated across, but not within pool pairs. This design was chosen to maximize family level diversity while maintaining an additional level of replication at the clutch pair level, while at the same time balancing logistics of resources and spatial constraints within pools. Enclosure assignment was haphazard for each clutch pair. In total across 10 pools, I stocked 100 enclosures containing approximately 100 embryos each. Breeding began first at the two most southerly pools on 2 April 2008. 8 April 2008 was the last day that breeding began in a new pool. Given the proximity between each pair of pools (i.e., the complement of a woodland-roadside pairing), breeding began within a one-day period across all pairs. Stocking of enclosures occurred in a pairwise manner between each roadside and woodland pond. In this way, there were no systematic differences in the timing of collection or stocking. I used a dissecting stereoscopic microscope to estimate the developmental stage of each egg mass upon collection from the field. For each embryo cluster (n = 200) I estimated egg size from photographs. Specifically, I used ImageJ (Rasband, 1997–2012) to measure the two-dimensional surface area of 15 embryos per cluster represented by a best-fit ellipse (Brady, 2012). At the conclusion of the experiment, when all eggs had either hatched and reached feeding stage or died, I estimated survival, developmental stage (Gosner, 1960), and snout-vent length (SVL) at the container level. One enclosure was removed from analysis due to predator intrusion.

Chronic road salt exposure experiment

Of the 10 egg masses collected from each pool, nine were haphazardly assigned into groups of three (hereafter “clutch triad”). From each egg mass, three separate clusters—each of approximately 30 embryos—were separated out and haphazardly assigned by clutch triad to each of three road salt treatments. Each cluster (n = 270) was photographed and embryo size was estimated as described above. Each experimental unit comprised one clutch triad, the unique composition of which was maintained across treatments providing an additional random effect. In total, each of 10 pools contributed representatives from nine egg masses, grouped in clutches of three, and exposed to three treatments for a total of 90 experimental units (3 clutch triads per treatment × 3 treatments × 10 pools = 90). The three treatments (low, medium, high) reflected the range of specific conductance found in the study region. All treatments comprised aged and conditioned tap water. Road salt comprising at least 95% NaCl obtained from the Connecticut Department of Transportation was added to the medium and high treatments to achieve specific conductance values of 1,000 µS and 4,000 µS respectively, which approximate the mean and maximum values I detected in roadside pools in the study region. No road salt was added to the low treatment, which had a natural specific conductance of approximately 175 µS. Embryos were reared in 11.3 l plastic containers (42 × 28 × 18 cm) filled with approximately 8 l of assigned treatment water. Each container was fitted with mesh hardware cloth to suspend embryos in the water column off the bottom of the container. Containers were housed on tables in an outdoor facility under 50% shade cloth. Concentrations of roadsalt were maintained in response to evaporation and rain by adding lower or higher concentration treatment water respectively, and through routine, complete water changes. When all embryos hatched or died, I estimated survival, developmental stage, SVL, and prevalence of axial malformations (Bantle et al., 1991).

Acute road salt exposure experiment

Following assignment of embryo clusters to reciprocal transplant and chronic exposure experiments described above, egg masses were retained within collection containers and placed haphazardly within 150 L plastic wading pools that served as incubators, located in an outdoor enclosure covered with 50% shade cloth. When embryos reached developmental stage 18, I separated out approximately 45 embryos from each egg mass and placed them into a common garden environment consisting of a 150 L wading pool filled with aged, conditioned tap water. Embryos were grouped by origin such that each wading pool contained embryos from each of the 10 egg masses representing a single natural pool. Upon approaching feeding stage (Gosner developmental stage 25 (Gosner, 1960)), larvae were fed a mixture of 3:1 ground rabbit chow and fish flakes at a daily ration of 10% average body mass. When all individuals appeared to reach feeding stage, I exposed wood frog larvae to a series of five concentrations of road salt, ranging from 0.5 to 10.0 g/L. These concentrations were selected to represent previously documented acute exposure assays of the wood frog (Sanzo & Hecnar, 2006). From each pool, 10 larvae were randomly stocked into each treatment and replicated across four experimental units per treatment (10 pools × 5 concentrations × 4 replicates = 200 experimental units). A 5.1 L container (33 × 20 × 11 cm) filled with 4 L of treatment water comprised each experimental unit. Larvae in each experimental unit were given a four-day food ration at the start of the experiment. After 96 h of exposure, I evaluated larval survival. In order to standardize developmental stage of larvae in the experiment, exposure was conducted in two separate, consecutive 96-hour rounds—the first comprising six pools (representing three pool pairs) and the second comprising four pair-wise pools (representing two pool pairs).

Population size and female fecundity

In spring 2008 following the conclusion of natural oviposition, I conducted visual counts of wood frog egg masses in each of the 10 pools to estimate population size of breeding females. In spring 2010, I collected 56 pairs of inbound wood frog adults from a subset of six pools (three roadside, three woodland) and haphazardly assigned male–female combinations grouped by origin pool. Pairs were placed in 11.3 L plastic containers filled with 2.5 L of aged spring water, equipped with a small branch and one white oak leaf (Quercus alba) to provide a substrate for oviposition. Containers were positioned on a slight angle to create a depth cline, and adults were allowed to amplex and oviposit. Following oviposition, egg masses were photographed and fecundity was estimated from each photo using ImageJ (Rasband, 1997–2012) to manually count the total number of eggs oviposited per female.

Characterizing the environment

In each pool, I measured seven environmental characteristics associated with amphibian distribution and performance (Wellborn, Skelly & Werner, 1996). Specific conductance, dissolved oxygen, pH, and wetland depth were measured once during the experiment, while temperature was measured every thirty minutes using deployed temperature loggers. Global site factor—a measure of solar radiation reaching the pool—was calculated from hemispherical photographs, while wetland area was estimated from visual rangefinder measurements. Specific methods are reported elsewhere (Brady, 2012). I also collected water samples at eight pools (four roadside, four woodland) to assay the concentration of chloride ions using liquid chromatography.

Statistical analyses

All statistical analyses were conducted in R V. 2.15.0 (R Development Core Team, 2012). I employed AIC based selection procedures to compose a suite of mixed effects models to evaluate performance variables across the G × E interaction both for transplant and exposure experiments. I analyzed survival and malformation each as bivariate responses of successes and failures. Growth and developmental rates were derived as exponential functions of change in size and stage, respectively, in relation to the number of days elapsed (e.g., [ln(final size)− ln(initial size)]/period). Initial size was defined as embryo diameter derived from estimates of embryo area, while final size comprised hatchling snout-vent length (SVL). These derived rates were characterized by Gaussian distributions in the statistical models. All models of performance variables were composed with and without embryo size as a covariate in order to estimate the potential influence of egg size mediated maternal effects (Laugen, Laurila & Merilä, 2002). Each response variable was evaluated across a suite of models that differed in random effects structure (see Tables S1–S3). For each response variable, I evaluated for inference the most parsimonious model with respect to both the significance of interacting main effects and the random effect structure as indicated by lowest AIC score (see Tables S4–S6). I also used mixed effects models to evaluate the influence of pool type on embryo size and fecundity. I used a combination of MCMC randomization methods and log-likelihood approaches to conduct inference (Bolker et al., 2009). I used MANOVA to evaluate the suite of abiotic variables characterizing the roadside and woodland environment. I used a standard linear model to evaluate the influence of pool type on population size, which was scaled to pool area and log-transformed to meet model assumptions. The full details regarding these analyses are described in the Text S1. Data for this study are deposited in the Dryad Repository: http://doi.org/10.5061/dryad.fb8tk.

Results

Within the field experiment, I found that regardless of the pool environment in which wood frog embryos were reared, the average survival of the roadside deme was lower than that of the woodland deme (Posterior mean = −0.964; 95% HPD = −1.492 to −0.454; Pmcmc = 0.001). Specifically, survival of embryos originating from the roadside deme averaged 73%, as compared to 86% for the woodland deme (Fig. 2). Thus, across both environments, survival of the roadside deme was 15% lower than that of the woodland deme. I found analogous outcomes when wood frog embryos and larvae were exposed to road salt, a predominant contaminant in roadside pools. Exposure to road salt throughout embryonic development resulted in an increased proportion of hatchlings with axial malformations (χ2 = 11.95, df = 8, 2, P = 0.003). At the highest concentration of road salt—corresponding to the maximum detected in roadside pools in this region—49% of roadside hatchlings developed axial malformations, as compared to 33% for woodland hatchlings (Fig. 3A); thus, axial malformations were on average 50% more prevalent in the roadside deme. In the acute exposure experiment across a series of five concentrations of roadsalt, survival by the roadside deme averaged just 56% compared to 72% for the woodland deme (Fig. 3B), yielding a 22% relative reduction in survival (Posterior mean = −7.569; 95% HPD = −14.517 to −0.508; Pmcmc = 0.038). No responses were influenced by embryo size (all P > 0.22, Tables S4 and S5). Similarly, embryo size did not differ across deme (MCMC mean = 0.0017; 95% HPD = −0.0006–0.0041; Pmcmc = 0.127). Despite a suite of negative consequences of road adjacency and road salt exposure (Fig. 3; see also Text S1 and Figs. S1 and S2), I found no difference in the population size of breeding females between roadside and woodland pools (F = 0.033, df = 8, 1, P = 0.861). However, female size and origin interacted such that the number of eggs provisioned by roadside as compared to woodland females increased at a steeper rate with respect to body size (Posterior mean = 21.856 95% HPD = 3.763–39.380; Pmcmc = 0.017). Across all populations, female length averaged 53.4 mm SVL. At this size, females from the roadside deme lay 10.5% more eggs than females from the woodland deme (Fig. 4). With regard to abiotic pool conditions, only specific conductance varied with respect to environment (Table S7), reflecting the input of chloride ions from road runoff. There was weak evidence that dissolved oxygen varied between the two environments (T = −2.19, df = 8, 1, P = 0.060), with roadside pools averaging 2.38 mg l−1 (±0.108 SE) as compared to 2.632 mg l−1 (±0.043 SE) in woodland pools. Roadside pools contained an average chloride concentration of 147 mg l−1 as compared to 3.4 mg l−1 in woodland pools, contributing to the 26-fold increase in specific conductance found in roadside pools (Fig. 1 bar graph).

Figure 2 Embryonic Rana sylvatica survival across the G × E interaction.

Survival (±1 SE) is shown here as the mean proportion of individuals surviving to feeding stage across all experimental units (N = 99). The woodland deme is represented by open circles while the roadside deme is represented by filled squares. The environment in which the animals were grown out is on the x-axis.

Figure 3 Malformations and survival following road salt exposure.

Rana sylvatica responses to experimental road salt exposure. Values (±1 SE) represent means of container level responses. (A) Proportion of embryos surviving to hatchling with axial malformations following chronic exposure to three concentrations of road salt: Low = 175 µS; medium = 1,000 µS; high = 4,000 µS (n = 90). Open bars represent woodland deme; filled bars represent roadside deme. (B) Proportion of hatchling survival following acute exposure to road salt (n = 200). Open circles represent woodland deme; filled squares represent roadside deme. Concentrations of total roadsalt (g/L) are shown. From lowest to highest, these concentrations yielded specific conductance values of 1,300, 6,430, 9,370, 13,150, and 17,830 µS, respectively.

Figure 4 Wood frog fecundity in relation to female size and deme.

Female Rana sylvatica fecundity. Number of eggs laid per female (n = 56) is shown in relation to female body size (snout-vent length [SVL]). Open circles represent females from the woodland deme; filled squares represent females from the roadside deme.

Discussion

Despite strong negative effects associated with the roadside environment, I found no evidence of local adaptation, but instead found evidence for maladaptive performance. Specifically, compared to the woodland deme, the roadside deme survived at lower rates, both in the roadside environment and when experimentally exposed to road salt; similarly, the roadside deme accrued more malformations following road salt exposure. Strikingly, the presence of a maladaptive response was not limited to these roadside contexts—even when reared in woodland pools, survival of the roadside deme remained lower than that of the woodland deme (Fig. 2). This suggests that the maladaptive performance induced by the roadside environment further spurs a generalized depression effect on the roadside deme, which manifests independently of the immediate negative influences of both the roadside environment and road salt exposure.

Because adult wood frogs exhibit high site fidelity (Berven, 1990) and have shown phenotypic differentiation across microgeographic environmental gradients (Skelly, 2004), I assumed that the embryos I collected represent a lineage of wood frogs specific to the collection pool (i.e., ‘population’). While I later discuss the implications of relaxing this assumption, for now I assume that the offspring entered into this experiment represent an independent population potentially connected by some level of dispersal and gene flow. With this in mind, the pattern of maladaptive performance seen here suggests that negative effects associated with exposure to the roadside environment are transmitted from parent to offspring prior to birth. In this way, embryos may be predisposed to reduced performance prior to any direct encounter with the roadside environment. Alternatively, maladaptive performance may have been caused by early embryonic exposure to roadside pools. Specifically, the first 36 h following natural oviposition but preceding collection—during which time embryos were exposed to their natal environment—may have had strong negative effects on embryos. Though such early exposure alone could explain the putative maladaptive response, preliminary evidence suggests that early exposure has no such carryover effect in this system (SP Brady, unpublished data).

In addition to any potential effect caused by early exposure, the overall pattern of parentally mediated maladaptive performance described here is consistent with several potential mechanisms, including maladaptation, non-genetic inherited effects (e.g., maternal effects), and demographic processes such as phenotype-biased habitat-oriented dispersal. Maladaptation can be defined most generally as a deviation from an adaptive peak caused by genetic processes (Crespi, 2000). As an empirical phenomenon, maladaptation is less common than local adaptation (Crespi, 2000; Spitzer, 2006). Yet even in reciprocal transplant studies where local adaptation was hypothesized and evaluated, maladaptation frequently occurs. In a review of this literature, Hereford (2009) found the frequency of maladaptation to be 0.29, as compared to the frequency of local adaptation, which was 0.71. Where it has been described, maladaptation at the population level is typically the consequence of migration load (Hanski, Mononen & Ovaskainen, 2011), resulting when asymmetrical gene flow from non-adapted source populations yields genotypes of sub-optimal fitness in the recipient population (Garcia-Ramos & Kirkpatrick, 1997). In the present study however, given the relative survival advantage of the woodland deme of wood frogs, regardless of environment, it seems unlikely that asymmetrical gene flow (i.e., from the woodland deme to the roadside deme) is limiting the local adaptation capacity of the roadside deme. If this were the case, we would expect similar phenotypes between populations grown out within the roadside environment. Likewise, phenotypic homogeneity would be expected if we view maladaptation through the lens of the adaptive phenotypic landscapes framework (Simpson, 1944), which predicts that maladaptation is a ubiquitous outcome of dynamic environments because the response to selection always lags one generation behind selection itself; therefore, the adaptive high point is constantly shifting before phenotypes can reach their adaptive peak (Crespi, 2000). However, this process alone does not explain the maladaptation seen here. Specifically, if we assume that the roadside deme did not differ from the woodland deme prior to the recent influence of roads, then we would again expect comparable performance declines between populations when exposed to the roadside environment.

In addition to the potential role for genetic based maladaptation, non-genetic processes may be responsible for the observed maladaptive performance of the roadside deme. For example, maternal effects, broadly defined as “a direct effect of a parent’s phenotype on the phenotype of its offspring” (Bernardo, 1996) may be associated with the roadside deme’s reduced ability to survive. In this system, if we assume that parents were born in a roadside pool, maternal effects could originate from their exposure to roadside water during their period of aquatic development, which spans newly laid embryos through metamorphosis. Similarly, aspects of the terrestrial roadside environment might propagate negative maternal effects. Yet, I found no evidence that egg size—a maternally mediated trait known to influence amphibian performance (reviewed by Mousseau & Fox, 1998)—varied across deme or had any effect on performance variables. Still, maternal effects in this system may be mediated by mechanisms independent of size. For example, nutrient availability and/or allocation of resources to egg quality may differ between these two environments. Alternatively, roadside embryos might accrue chemical contaminants as in utero eggs through maternal transfer. This possibility is supported by outcomes from studies in which coal combustion waste contaminants such as strontium and selenium are maternally transferred, negatively affecting offspring survival and development in two species of toads (Hopkins et al., 2006; Metts et al., 2012).

Demographic processes might also lead to maladaptive performance of the roadside deme. For example, if we assume that roadside adults face higher mortality risk—either through carryover effects from the aquatic environment, or through terrestrial road pressures such as roadkill—then the average age of roadside breeders should be lower, a condition associated with poorer offspring provisioning in wood frogs (Berven, 1988). On the other hand, demographic processes might also explain how the roadside deme can persist in spite of the negative effects of the roadside environment and given a predisposition for lower survival. Specifically, across demes, I detected no difference in population size of breeding females. As one possible explanation of this demographic pattern, I found that on average, and adjusted for mean body size, females from the roadside deme lay 10.5% more eggs than females from the woodland deme. Thus while average survival is 15% lower for roadside embryos, the differential in numerical reproductive output per female is less substantial. For example, based on the overall average female size, roadside wood frogs lay 890 eggs per clutch while woodland wood frogs lay 795 eggs per clutch. If we apply the deme respective survival rates (73% versus 86% respectively), we can estimate that for the average sized female, 684 embryos survive to feeding stage in the woodland deme as compared to 650 embryos in the roadside deme. This amounts to just a 5% decrease in reproductive success per female of the roadside deme. Thus, differential reproductive efforts may potentially offset average reductions in offspring survival, mitigating the detriment to demographic success. This increase in reproductive effort may serve to buffer the population against stochastic effects in the face of intense of selection. Ultimately, given the interacting effects of female size and deme (Fig. 4), the adaptive significance of this increased reproductive effort will depend upon the distribution of female size. Similarly, the potential for a tradeoff between quality and quantity of eggs will affect the utility of this apparent strategy.

A second demographic possibility is that roadside pools may serve as sink habitats, wherein roadside populations are augmented through immigration from woodland populations, bolstering the chance of persistence. However, here again, if this were true, we might expect that the migration of woodland individuals into roadside pools would yield homogenized performance responses between the roadside and woodland deme, which is inconsistent with the observed pattern. Still, source–sink dynamics could occur if we assume that putative migrants arriving to breed in roadside pools are themselves depressed in some way. For example, younger or otherwise misfit adults may breed more frequently in the suboptimal conditions found in roadside as compared to woodland pools, potentially through competitive exclusions or the process of ‘matching habitat choice’ (Edelaar, Siepielski & Clobert, 2008). As an alternative explanation of persistence, it is possible that the negative effects on the aquatic stages of roadside wood frogs might be reversed at later life history stages. For example, the surviving individuals from the roadside deme may produce inferior embryos and larvae, but superior adults.

It is surprising to consider that the maladaptive response of the roadside deme was unabated in woodland pools, where survival is typically high. While this generalized depression effect (hereafter ‘deme depression’) is not well described in the literature (but see Falk et al., 2012), there are conceptual analogues worth considering. As one such example, inbreeding depression can arise in the context of reduced effective population size (Newman & Pilson, 1997) that could result from intense selection (Robertson, 1961) in the roadside environment. However, inbreeding depression is predicted to be minimal in the context of intense maladaptation (Ronce et al., 2009). Still, roads can act as barriers to migration, which can potentially spur inbreeding depression. However, the types of roads in this system do not appear to limit gene flow between wood frog populations in surrounding landscapes separated by similar or larger distances of space (Richardson, 2012). A similar argument for gene flow could therefore be made against genetic drift as a mechanism driving these patterns. As an alternative possibility, genetic mutations or epigenetic modifications could lead to maladaptive responses and overall deme depression. This may be especially likely in harsh roadside environments, where mutagens such as heavy metals are known to accumulate from road runoff (Trombulak & Frissell, 2000).

Regardless of mechanism, the maladaptive performance of the roadside deme of wood frogs occurs in direct contrast to the local adaptation of the roadside deme of spotted salamanders breeding and dwelling in the very same pools (Brady, 2012). That deme divergence has been detected for two different species occupying identical habitats suggests that population differentiation may be a common consequence of landscape modification. That these outcomes occurred in opposing directions highlights the complex nature of population level responses across species—even when they occupy identical habitats—suggesting that more generalized approaches may be inadequate, and reinforcing the imperative for population specific investigations. Though microgeographic population outcomes have received little attention, there is reason to believe that maladaptive responses may be an emerging and prominent consequence of human environmental modification. For example, phenotypes of species harvested for human consumption have undergone rapid and substantial change (Darimont et al., 2009) in directions believed to oppose adaptation to the increased environmental variation predicted to occur with climate change (Koons, 2009). Similarly, hatchery-reared fish have become locally adapted to the hatchery environment at the cost of maladaptation to the wild (Christie et al., 2012). It is worth noting how the conservation implications of these cited examples of maladaptation differ from that of the maladaptive responses reported here. Wherein harvest regimes and hatcheries induce maladaptation through selection as a consequence of a specific management strategy, roads induce maladaptive performance as a collateral effect of global change and for a species that is a bystander to human actions, not a direct target. Thus, even indirect human activities in relatively undeveloped settings can induce maladaptive responses in wild populations.

Supplemental Information

Text S1 Supplemental Text

Click here for additional data file.

Supplemental Information S2 Supplemental Methods and Statistical Analyses

Click here for additional data file.

Figure S1 Overview of reciprocal transplant experimental methods.

Procedural steps for stocking reciprocal transplant enclosures. The reciprocal transplant was conducted pairwise across ponds. Specifically, each of five roadside pools was paired with each of five nearby woodland pools; the transplantation of embryos occurred within each of these five pairs (see main text and Fig. 1). Text color corresponds to indicator arrows and ellipses. From top to bottom: 10 wood frog egg masses were collected from each of 10 pools (Step 1). A breeding pair of wood frogs is shown; red circles indicate the approximate extent of individual egg masses. “Clutch pairs” were formed by subsampling and combining ca. 50 embryos from each of two masses per population (Steps 2, 3, 5). Photographs were captured with a scale in view to allow estimation of embryo size (Step 4). Replicate cutch pairs were stocked into natal and transplant pools respectively; enclosures are shown. (Step 6).

Click here for additional data file.

Figure S2 Population level embryonic survival across the G × E interaction

Wood frog survival at the population level shown across the genotype × environment (G × E) interaction. Open symbols (connected by dashed lines) represent woodland populations; closed symbols (connected by solid lines) represent roadside populations. Like colors and symbols represent paired pools across which transplants were conducted. Hatchling survival (n = 99 was assessed when animals reached feeding stage. Deme level responses (i.e., averaged across populations) can be found in the main text (Fig. 2) and reproduced below (Fig. S3, panel a). Values (±1 SE) represent population level means of enclosure level responses.

Click here for additional data file.

Figure S3 Wood frog survival, growth, and development across the G × E interaction.

Wood frog performance across genotype x environment (G × E) interaction. Hatchling survival (n = 99), growth, and development (each n = 95) were assessed when animals reached feeding stage. Values (±1 SE) represent means of enclosure level responses. Panel ‘a’ is reproduced here from the main article for reference.

Click here for additional data file.

Figure S4 Wood frog performance following chronic exposure to road salt.

Wood frog performance following chronic exposure to road salt. Hatchling survival (a), growth (b), and development (c; each n = 90) were assessed when all animals hatched from eggs or died. Values (±1 SE) represent means of container level responses. There was no evidence that road salt affected these performance variables over the period of embryonic development. Solid bars represent the roadside deme; open bars represent the woodland deme.

Click here for additional data file.

Table S1 Reciprocal transplant model selection results

Reciprocal transplant model selection results. A set of candidate models differing in random effects structure was composed for the analysis of each response variable. An observation level term (“obs”) was included to test and account for over-dispersion in the binomial model of survival. The model containing the fewest parameters with the lowest Akaike Information Criterion (AIC) score by a differential value of less than two was chosen for inference, and is indicated by a dagger (†). There was no significant genotype x environment (G x E) interaction in the model for survival, growth rate, or developmental rate.

Click here for additional data file.

Table S2 Chronic exposure model selection results

Chronic exposure model selection results. Candidate models were composed for survival, growth, development, and malformations across the interaction of genotype (G) x environment (E); here environment refers to the three different road salt treatments. Model selection is described in caption for Table S1. An observation level term (“obs”) was included to test and account for over-dispersion in the binomial models. There was no significant G x E interaction in the model for survival, growth rate, or developmental rate.

Click here for additional data file.

Table S3 Acute exposure model selection results

Table S3. Acute exposure model selection results. Candidate models were composed for survival across the interaction of genotype (G) x environment (E); here environment refers to the five different road salt treatments. The model with the fewest parameters and lowest Akaike Information Criterion (AIC) score by a differential value of less than two was inferred, and is indicated by a dagger (†). Experiment block was nested within experiment round (“rnd”) because it was assumed that potential blocking effects would not be independent of experiment round. An observation level term (“obs”) was included to test and account for over-dispersion.

Click here for additional data file.

Table S4 Parameter estimates, confidence intervals and p-values of field transplant models selected for inference

Parameter estimates, confidence intervals and p-values of field transplant models selected for inference. All models were composed with and without egg size as a covariate.

Click here for additional data file.

Table S5 Contrasts of G × E interaction on prevalence of malformations

Contrasts of G × E interaction on prevalence of malformations. Environment (E) refers to the three different road salt treatments. The model was composed with the original interaction effect of G × Ereformulated as one main effect (referred to as G × E†) comprising six levels (two demes X three treatments). Three contrasts were selected to infer responses between demes (R = roadside; W = woodland) within each treatment (L = low; M = medium; H = high). The remaining two orthogonal contrasts tested for differences between treatments irrespective of deme.

Click here for additional data file.

Table S6 Parameter estimates, confidence intervals and p-values of acute exposure model selected for inference

Parameter estimates, confidence intervals and p-values of acute exposure model selected for inference. The interaction term was marginally significant and so was retained in the model. Embryo size could not be included in the model because clutch groupings were not separately maintained in the experiment.

Click here for additional data file.

Table S7 Parameter estimates of environmental variables

Parameter estimates of environmental variables. Estimates were obtained from univariate ANOVAs on each environmental variable following an overall significant MANOVA (Pillai’s Trace = 1.00, df = 8, 1, P = 0.016). For each variable, estimates are given for both the intercept and the pool type, the latter showing the effect with respect to roadside pools designated as the reference level. Specific conductance and pool area were log-transformed to improve normality. The multivariate analysis was conducted in R using the function manova, while univariate analyses were conducted using the function lm, both of which are in the Base Package.

Click here for additional data file.

I am grateful to the Leopold Schepp Foundation for their support of my work through scholarship. I thank D Skelly, S Alonzo, P Turner, M Urban, A Hendry, and R Calsbeek for key guidance and project advice. I am grateful to A Brady and S Bolden for extensive support. I thank the reviewers for their constructive and keen feedback on the original manuscript. I thank D Cholewiak, D Risch, and D Sigourney for helpful discussions. S Attwood, J Burmeister, G Antonioli, J Richardson, M Manickam, M Rogalski, H McMillan all assisted in the field. Z, M, and A Ladzinkski and S and R Brady greatly assisted with infrastructure. J Bushey provided chemical analysis. The Connecticut Department of Transportation provided road salt for exposure assays.

Additional Information and Declarations

Competing Interests

Author Contributions

Animal Ethics

Field Study Permissions

Data Deposition

The author declares no competing interests.

Steven P. Brady conceived and designed the experiments, performed the experiments, analyzed the data, contributed reagents/materials/analysis tools, wrote the paper.

The following information was supplied relating to ethical approvals (i.e., approving body and any reference numbers):

Yale University Institutional Animal Care and Use, project numbers 2006-11024 and 2009-11024.

The following information was supplied relating to field study permissions (i.e., approving body and any reference numbers):

Animal collections were permitted by the State of Connecticut Department of Environmental Protection. Permission to use study sites was granted by the Yale School Forests and by private land-owners.

Data for this study are deposited in the Dryad Repository: http://doi.org/10.5061/dryad.fb8tk.

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
