# Peer review of "Microgeographic maladaptive performance and deme depression in response to roads and runoff"

_PeerJ, doi:10.7717/peerj.163_

## Round 0.1 · original submission · Major Revisions

The three referees found this manuscript interesting and timely, and I agree with them. The experimental design and the statistical analyses are sound, even though the referees asked for more detailed information, a more accurate interpretation and a clarification of terms.

·

Basic reporting

The manuscript by Steven Brady about local adaptation to relatively new habitat like roadside pools in amphibians is a nice example of the effects and consequences of landscape alteration of humans and the apparently incapacity of many organisms to adjust or adapt correctly to these new environments. In my opinion is a good contribution to conservation ecology from an evolutionary point of view.
I will like but to suggest to the author to change the title, and remove the word "fragmented landscape". I agree with him, that roads (which is the main element analyzed in this manuscript as factor), contribute to landscape fragmentation, but from my point of view, the manuscript is not so much about "strict" fragmentation concept (reduction of habitat in small spots,etc.) but about new "human created" habitats. The forest pools where frogs are breeding are not reduced their areas, and the authors considered that gene flow among ponds is not affected by roads and traffic. The main factor is that pools closer to roads showed a different chemistry composition that those apart from the roads. I think that maladaptation is a consequence of contamination or alteration of habitats but not of landscape fragmentation.

Experimental design

Field transplant experiments are nicely complemented with laboratory experiment where different factors like salt concentration can be isolated to test the effects of the main factor which could be involved on the adaptation or not of demes from different environments. The number of demes (10) distributed in five pair (roadside - forest), the number of clutches used from each environment and number of eggs and individuals used in each experiment, as the number of replicas give guarantees for the robustness of the results and acceptance or rejection of hypothesis.

Validity of the findings

Statistical analyses are also rightly performed, using in all cases good statistical approaches to analyze the data obtained from the field and laboratory experiment. The control of effect by egg size, or mother size allow the author to control potential maternal effects which could difficult the interpretation of the results obtained.
The figures are very compressible and help to readers to get a clear image of the results. I will only suggest to the author to change the scale of some figures, specially the one related with salt concentration treatments. Instead of using same distance between treatments I will like to use a lineal scale in the X-axis to better visualize the response of demes to different salt concentration treatment. Using a lineal scale it would be easier to see if mortality increase is lineal or there is some point of inflexion at which mortality increase in a exponential way.

Additional comments

I will suggest to the author to discuss a little bit about the differences in clutch size between roadside and forest demes. In the results it seems that it is a statistical difference between them, with females from roadside demes producing more eggs than forest demes: around 10% more eggs. Considering that mortality from these demes in higher: 15% in transplant experiments, it can be that the increase of egg deposition in these demes would be an adaptive response to "compensate" the higher embryos mortality. I think that this is an important point to be discussed in the discussion section, as a mechanisms that allow to roadside demes to survive and persist in those environments even in depress conditions for longer time before they completely disappear or adapt to the new conditions.

·

Basic reporting

No comments

Experimental design

I’ve found quite difficult to follow the design of the reciprocal transplant experiment. A scheme with the design in the supplementary file would be very helpful to understand the allocation and levels of replication.
The data is not available for review. The author will make them available when accepted for publication.
Figures and results for survival are summarizing means and SEs for the different origins/demes (roadside vs woodland). I would appreciate to see the results (and figures if it’s not too messy) broken down at the experimental pool level to see whether the lower performance of the roadside origin is consistent across pools.
It is important to have more accurate information about the specific timing of egg collection in each deme and the stocking. Any potential differences in breeding time, especially between roadside and woodland pools, may have an impact on the results (implications explained in the “validity of the findings” section.
The full results of the analyses of the abiotic variables should be presented MANOVA table and descriptive stats (at least in the supplementary file). No doubt the author only found differences in conductance, but it would be informative to see the rest as well. Without knowing the pool first hand, one would expect differences in vegetal cover/radiation and probably temperature between woodland and roadsides.
Info about the specific R packages and procedures used to run the GLMMs would be appreciated.
Many people may not be familiar with Bayesian inference and it would be good clarifying what the posterior mean refers to (eg. line 197) and the mean of what is being reported. Since the posterior mean is reported then is appropriate to give the associated 95% HPDI, not CIs, also explanation about how the Pmcmc is estimated would be helpful to grasp differences with frequentist p-values.

Validity of the findings

I find this study very relevant. Studies of microevolutionary (or lack of thereof) intraspecific processes at small scale are much needed. This is a good example of a pertinent well carried study with appropriated well-supported discussion.
Some minor comments:
- Differences in the onset of breeding (ca. days) determine growth opportunities and force acceleration/compensation in growth/development rates which, in turn, may have a cost in life-history traits (i.e. condition, size at metamorphosis, depression of immune response). For that reason would be important knowing if there was any significant difference in the times of collection and stocking of the clutches in the experimental pools. Especially any systematic difference that may have occurred between roadside and woodland pools. Perhaps this is not an issue in this study but it would be important to clarify it.
-Lines 228-230. Perhaps “inherited” is not the best word choice since, as treated extensively in the discussion, there are many non-genetic potential causes behind the observed pattern.
-Lines 265-270. In addition to accumulation of harmful chemicals may be other components of yolk quality (independent of egg size) differing between origins.
-Lines 278-283. Assuming equal quality of the eggs of the two origins (woodland vs roadside) I don’t follow this reasoning. The demographic success will rely on the actual number of eggs but not on the number of eggs relative to parent size. If without adjusting for body size there is no difference in fecundity between origins, no differential demographic effects would be expected (everything else being equal).

·

Basic reporting

Pass

Experimental design

Needs some clarification - see comments to author.

Validity of the findings

Good - but needs some improvement of interpretation - see comments to author.

Additional comments

This paper describes an interesting study that tested for local adaptation (woodland versus roadside ponds) by salamanders. The authors perform the usual local adaptation experiments through reciprocal transplants in nature and environmental manipulations (roadside salt) in the lab. They also estimate population sizes in the two environments in nature. Of particular interest, the authors find an unexpected pattern of maladaptation by the roadside salamanders – they show lower fitness under all conditions. This result appears quite strong and robust and is therefore very interesting and worth publishing.
Comments:
1. Lines 38-43: Somewhere you need to say what the likely selective factor is – salt. You can’t leave the reader guessing for so long.
2. Line 76: Wouldn’t unpaved roads also be salted?
3. Line 80: But it would introduce a bias in the sense that eggs exposed to salt might be affected in such a way as to cause later reductions in survival even when transferred to non-salt. Thus, it is possible that this 36 hour differential exposure is the reason for the results – rather than local adaptation, differential migration, or maternal effects.
4. Line 238: The dispersal would have to also be phenotype biased, which might not be clear based on the statement here. Also, the 36 hours of exposure before collection is another possibility.
5. Line 240: I am not sure what is meant by “rarely reported.” Hereford et al. show many instances where performance is not higher in local environments. (I am also reminded of the Hendry and Gonzalez’s Biology and Philosophy debate about adaptation.) Maybe formally compare values from the current study to those in Hereford. For instance, how many studies in that meta-analysis fall into the category of one type performing better in both environments and, within that category, how strong is your result compared to the others?
6. For maternal effects to be the cause, the mothers that breed in roadside ponds would also have to have been born in those ponds, right? Or do maternal effects accrue during the short period a female sits in the pond before laying? In short, be more clear on just how the maternal effects are expected to work.
7. Line 297-304: See Falk et al. (2012) in EER for a possible example.
8. Roadside females are said to have higher fecundity than woodland females but inspection of the data (Figure 4) shows that this is true only for large females. When making this sort of test, a significant interaction (I can’t find information on whether it is or is not significant) means that one can’t make blanket statements about the difference in egg size – because it depends on female size. (Note that this is not fixed by “adjusting for body size.”)
I found the writing to be rather obtuse, with many abstract phrases and terms for which the meaning is ambiguous. The author should strive to improve clarity and to use more straightforward English whenever possible. Here I list a number of those instances – and these and others should be clarified. In fact, the entire MS should be examined for such instances.
1. Abstract: define “respond maladaptively”
2. Line 16: what is meant by “trajectories”?
3. Line 19: define “deme” – “population” is used more commonly (although I realize deme is the classical pop gen version) and it may not be clear if they mean similar or different things. Moreover, it is confusing later how this term is used. For instance, line 196 (and line 207 and elsewhere) refers to THE roadside deme but there are five roadside ponds, so doesn’t that mean there could be five roadside demes. If by deme, you mean reproductively isolated population, then you don’t know which are demes or not since you don’t have genetic data reported. In short, this is all confusing and needs improvement.
4. Line 19: Only a specific type of GxE is consistent with local adaptation.
5. Line 67: Might not be clear to some whether “site selection” is by yourself or by the amphibians. Also it is not clear what is meant by “8 of 10 overlapping pools” – what happened to the other two and how are they “overlapping”?
6. Line 85: Not clear what is meant by “weather conditions.” How is this relevant?
7. Line 131: Confusingly worded as “dissected” for most people means the embryos are cut up, in which case they would survive. I assume you mean individual eggs are separated but you should clarify this.
8. Line 146: I am not sure it is clear why the pools are “pair-wise”
9. Line 177: The use of diameter here seems to conflict with the “two-dimensional surface area” mentioned earlier.
10. Line 215: adjusting for body size is important but this phrase wasn’t used in the methods and so I am not sure how the adjustment was done.
11. Line 224: selection wasn’t measured – perhaps rephrase
12. Line 240: Preferentially means they prefer the site but some of the mechanisms here are instead competitive displacement, not preference.

---

## Round 0.2 · accepted · Accept

All the points raised by the three referees have been satisfactorely addressed in the revised version of the manuscript. Thank you for this nice piece of work.